# Metabolic Features of Increased Gut Permeability, Inflammation, and Altered Energy Metabolism Distinguish Agricultural Workers at Risk for Mesoamerican Nephropathy

**DOI:** 10.3390/metabo13030325

**Published:** 2023-02-22

**Authors:** Nathan H. Raines, Dominick A. Leone, Cristina O’Callaghan-Gordo, Oriana Ramirez-Rubio, Juan José Amador, Damaris Lopez Pilarte, Iris S. Delgado, Jessica H. Leibler, Nieves Embade, Rubén Gil-Redondo, Chiara Bruzzone, Maider Bizkarguenaga, Madeleine K. Scammell, Samir M. Parikh, Oscar Millet, Daniel R. Brooks, David J. Friedman

**Affiliations:** 1Division of Nephrology, Department of Medicine, Beth Israel Deaconess Medical Center, Harvard Medical School, Boston, MA 02215, USA; 2Department of Epidemiology, Boston University School of Public Health, Boston, MA 02118, USA; 3Faculty of Health Sciences, Universitat Oberta de Catalunya, 08018 Barcelona, Spain; 4ISGlobal, Barcelona Institute for Global Health, 08003 Barcelona, Spain; 5Universitat Pompeu Fabra (UPF), 08002 Barcelona, Spain; 6CIBER Epidemiología y Salud Pública (CIBERESP), 28029 Madrid, Spain; 7Department of Environmental Health, Boston University School of Public Health, Boston, MA 02118, USA; 8Precision Medicine and Metabolism Laboratory, CIC bioGUNE, Basque Research and Technology Alliance (BRTA), 48160 Derio, Spain; 9Division of Nephrology, Department of Medicine, University of Texas Southwestern Medical School, Dallas, TX 75390, USA; 10CIBERehd, Instituto de Salud Carlos III, 28029 Madrid, Spain

**Keywords:** Mesoamerican nephropathy (MeN), chronic kidney disease of unknown etiology (CKDu), chronic kidney disease of nontraditional cause (CKDnt), nicotinamide adenine dinucleotide (NAD+), kynurenate, tryptophan, hippurate, agriculture, manual labor

## Abstract

Mesoamerican nephropathy (MeN) is a form of chronic kidney disease found predominantly in young men in Mesoamerica. Strenuous agricultural labor is a consistent risk factor for MeN, but the pathophysiologic mechanism leading to disease is poorly understood. We compared the urine metabolome among men in Nicaragua engaged in sugarcane harvest and seed cutting (*n* = 117), a group at high risk for MeN, against three referents: Nicaraguans working less strenuous jobs at the same sugarcane plantations (*n* = 78); Nicaraguans performing non-agricultural work (*n* = 102); and agricultural workers in Spain (*n* = 78). Using proton nuclear magnetic resonance, we identified 136 metabolites among participants. Our non-hypothesis-based approach identified distinguishing urine metabolic features in the high-risk group, revealing increased levels of hippurate and other gut-derived metabolites and decreased metabolites related to central energy metabolism when compared to referent groups. Our complementary hypothesis-based approach, focused on nicotinamide adenine dinucleotide (NAD+) related metabolites, and revealed a higher kynurenate/tryptophan ratio in the high-risk group (*p* = 0.001), consistent with a heightened inflammatory state. Workers in high-risk occupations are distinguishable by urinary metabolic features that suggest increased gut permeability, inflammation, and altered energy metabolism. Further study is needed to explore the pathophysiologic implications of these findings.

## 1. Introduction

Mesoamerican nephropathy (MeN) is a devastating form of chronic kidney disease (CKD) in agricultural communities in the Pacific coastal region of Mesoamerica [1]. Two decades after its initial description, important questions remain about how environmental exposure and physiologic susceptibility intersect to produce kidney disease in at-risk populations. MeN is characterized as a non-diabetic non-hypertensive syndrome of CKD with minimal proteinuria, sterile pyuria, hyperuricemia, and nonspecific biopsy findings demonstrating tubular injury with ischemic features and cellular inflammation [2,3]. Notably, this description of MeN lacks details regarding the etiology and pathophysiology of the disease, as they remain poorly understood. 

Environmental stressors linked to agricultural labor are of particular interest in MeN, given the high prevalence of the disease among agricultural workers. Through use of metabolomics, the study of patterns of small molecules derived from cellular metabolism and exogenous sources [4], we can begin to understand the physiologic processes arising in response to exposures related to agricultural work [5,6]. 

An area of particular interest regarding the pathophysiology of MeN is derangement in the biosynthesis of nicotinamide adenine dinucleotide (NAD+). NAD+ is an essential coenzyme linking glycolysis and the citric acid (tricarboxylic acid, TCA) cycle with the mitochondrial electron transport chain [7,8]. NAD+ also participates in cellular repair and immune function [7,8]. Deficiency of NAD+ can become rate-limiting for normal oxidative metabolism, which in turn may impair organ function [7]. 

Alterations in energy metabolism centered around NAD+ emerge in the setting of physiologic stress in the kidney. These alterations are characterized by a decline in cellular NAD+ biosynthesis mediated in part by decreased activity of quinolinate phosphoribosyltransferase (QPRT) [9,10]. Depletion of cellular NAD+ plays a role in the pathophysiology of ischemic and inflammatory renal tubular injury [9,10,11,12]. While the contributions of ischemia and inflammation to the pathogenesis of MeN are not definitively established, they are among the leading hypothesized causes underlying the development of MeN [2,13,14]. NAD+ biosynthetic derangement and broader dysfunction in energy metabolism are also involved in the pathogenesis of chronic kidney disease, including fibrogenesis [15,16,17]. Exploration of NAD+ biosynthetic derangement among individuals engaged in activities associated with the development of MeN may therefore provide a link between these activities and disease pathogenesis. 

NAD+ biosynthesis occurs along three key pathways: the de novo pathway that utilizes tryptophan (Trp), the salvage pathway that uses nicotinamide (Nam), and the Preiss–Handler pathway that uses the acid form of Nam, nicotinic acid (Figure 1) [7]. In vivo, measurement of component metabolites in these pathways can provide insight into metabolic activity. For example, urine kynurenic acid (KynA), a derivative of Trp, is a useful biomarker of inflammation due to its role as a modulator of inflammatory response [18]. Increased urinary KynA levels are linked to non-recovery from acute kidney injury (AKI) [19]. Urine quinolinate to Trp (Q/T) ratio can be used as a biomarker of NAD+ biosynthetic derangement, with elevated ratios associated with kidney injury [10].

In this study, we used proton nuclear magnetic resonance (^1^H NMR) to characterize the urinary metabolic features of men in Nicaragua working as sugarcane harvest or seed cutters, job categories at very high risk for MeN based on prior research [20,21,22]. We compared this extreme risk group against three referent groups: Nicaraguan men who worked in sugarcane but not as sugarcane harvest or seed cutters; Nicaraguan men not working in agriculture; and men in Spain working in non-sugarcane agriculture. All individuals in this study had preserved kidney function, as evaluated by creatinine-based estimated glomerular filtration rate (eGFR) in order to minimize the impact of altered kidney function on urinary metabolite levels. In the primary analysis, we employed a non-hypothesis-based approach to evaluate differences in metabolic features between the high-risk sugarcane harvest/seed cutters and each referent group. In secondary analysis, we employed a complementary hypothesis-based approach, hypothesizing that NAD+ related metabolites would differ between the extreme risk group and referent groups.

## 2. Materials and Methods

Research protocols were approved by the institutional review boards (IRBs) of Boston University Medical Campus (BUMC; Boston, MA, USA), Beth Israel Deaconess Medical Center (BIDMC; Boston, MA, USA), and the Parc de Salut Mar ethics committee (Barcelona, Spain). Participants in each of the four component studies contributing specimens for this analysis provided individual informed consent for adjunct specimen analysis related to kidney disease.

### 2.1. Study Design

This is a cross sectional study using urine specimens from individuals in Nicaragua and Spain who were identified as having an eGFR > 75 mL/min/1.73 m^2^ based on the chronic kidney disease epidemiology consortium (CKD-EPI) 2021 serum creatinine equation without a race term, an equation shown to be accurate in MeN-affected populations [23,24]. Individual participants from the component studies in Nicaragua were selected to include only individuals where all available eGFR measurements were >75 mL/min/1.73 m^2^. Similar longitudinal creatinine data was not available for the Spanish population, so inclusion was based on a single eGFR measurement. The final sample included four distinct groups: Nicaraguan men working as sugarcane harvest or seed cutters (cut and seed, CS; *n* = 117); Nicaraguan men who work in sugarcane cultivation but not as sugarcane cutters (Other sugarcane workers, OW; *n* = 78); men who work in agriculture in Spain (Spanish workers, SW; *n* = 78); and Nicaraguan men who work in either the mining or brickmaking industry (non-agricultural workers, NW; *n* = 102).

We tested voided urine specimens collected at the end of a work shift for the CS, OW, and SW groups, and at varying times during the day for the NW group. The CS and OW group specimens were collected in 2015 during an ongoing case-control study investigating risk factors for MeN in Nicaragua, as well as during follow-up from a previously published prevalence study in Nicaragua [25]. Specimens comprising the SW group were collected in 2018–2020 during a cross-sectional study, the “Acute Kidney Injury in Agricultural workers in Spain: risk factors and long term effects” (LeRAgs; https://www.isglobal.org/ca/-/lerags, accessed on 1 October 2022) study, in three provinces in Spain in which agricultural workers were recruited to study the association between environmental/occupational exposures and kidney disease. Specimens comprising the NW group were collected in 2016 and came from two previously published studies: one a case-control study among individuals working in mining-related activities, and one a prospective cohort study among individuals working as brickmakers [26,27]. All specimens were stored frozen at −80 °C from the time of collection until analysis.

Pre-specified assessment of risk for each group is presented in Table 1. The CS group was considered highest risk, with each referent group considered to have lower but not necessarily equally low risk. Risk was categorized based on residence in a high-risk region; manual labor intensity, with more intense labor considered highest risk; and occupational environment, with agricultural work considered highest risk. 

### 2.2. Laboratory Methods

Urine specimens from all participants were analyzed using an AVANCE IIIHD (IVDr) 600 MHz Bruker NMR spectrometer equipped with a PA-BBI probe head with z-gradient coil and an automatic SampleJet sample changer operating at 300 K and using the standard operating procedures of the Phenome Center Consortium. For each sample, two different ^1^H NMR spectra were collected: a high-resolution 1D ^1^H spectrum to obtain quantitative metabolite data for statistical analysis and a 2D-Jres experiment for assistance in peak assignment and metabolite identification. Each spectrum was segmented into consecutive buckets (bins) of fixed 0.01 ppm width in the region between 9.5 and 0.5 ppm. Concentration for each metabolite was analyzed using the TSP signal in the urine spectra fitted to a generalized extreme value (GEV) distribution using a maximum—likelihood estimation (MLE) algorithm with Broyden–Fletcher–Goldfarb–Shanno (BFGS) optimization.

### 2.3. Data Analysis

Data on demographics, anthropometry, and basic laboratory features of each of the CS, OW, NW, and SW populations were collected during the respective parent studies. Information on age, sex, medication use, and diagnosis of hypertension and diabetes was obtained in parent studies using participant questionnaires. Procedures for laboratory analyses were reported in the parent publications for the CS, OW, and NW populations [25,26,27]. For the SW cohort, urinalysis was conducted by dipstick and serum creatinine was measured using the Jaffe method.

In non-hypothesis-based analyses, we compared levels of all measured metabolites between the CS and referent groups. We also analyzed levels of metabolites with groups pooled and compared by shared risk factors, as described in Table 1: Residence in a high-risk area (CS, OW, NW) or not (SW); work in higher (CS) versus lower (SW, NW, OW) physical intensity jobs; and work in agriculture (CS, OW, SW) or not (NW). Metabolites were analyzed both individually and grouped by metabolic pathway. Pathway analyses incorporated both differences in metabolite concentrations between groups and the impact of measured metabolites has on the overall activity of the pertinent metabolic pathway [28]. Pathways were defined using the Kyoto Encyclopedia of Genes and Genomes (KEGG) [29]. 

In our hypothesis-based approach, we compared urinary levels of Trp, KynA, NAD+, Nam, and 1-methylnicotinamide (MNA) to evaluate activity along the de novo and salvage pathways of NAD+ metabolism (Figure 1). To evaluate activity along different pathways of NAD+ biosynthesis, we compared ratios of concentrations of metabolites within each pathway. NAD+/Trp and KynA/Trp ratios were used as markers of activity favoring conversion of tryptophan to downstream metabolites, and NAD+/KynA as a marker of the relative concentrations of these downstream metabolites [19]. Similarly, NAD+/Nam and MNA/Nam ratios were used as markers of activity favoring conversion from Nam to downstream metabolites, and NAD+/MNA a marker of the relative concentration of downstream metabolites [30]. Quinolinate was not detected in the majority of samples using our ^1^H NMR methodology, so we were unable to evaluate the Q/T ratio as an indicator of NAD+ biosynthesis from Trp. Comparisons were made both across groups overall and between the CS and each referent group individually. We conducted a post-hoc analysis of the association between urine hippurate and KynA/Trp ratio. As a stability analysis, we repeated hypothesis-based analyses with the CS group divided into its component sugarcane harvest cutter and sugarcane seed cutter groups to determine whether differences exist between these two high-risk groups. We also repeated non-hypothesis-based analyses in the subgroup individuals with an eGFR ≥ 90 mL/min/1.73 m^2^.

### 2.4. Statistical Methods

All metabolite concentrations were normalized to urine creatinine. Two extreme value outliers driven by very low urine creatinine were removed prior to analysis. Missing values were assumed to be below the threshold for detection by ^1^H NMR, and were imputed as 1/5th the minimum positive value of that feature [31]. Metabolites with all concentration values below the threshold of detection were excluded from analysis.

Comparisons between groups in non-hypothesis-based analyses were made using the false discovery rate (FDR) statistic to account for multiple comparisons. Metabolite concentrations were log-transformed and mean-centered divided by the standard deviation to achieve normal distribution prior to analysis. We performed supervised partial least squares discriminant analysis (PLS-DA) and a random forest analysis with out-of-bag (OOB) error calculation to determine how consistently patterns of metabolites were able to identify group alignment, and to determine important metabolites in creating this identifying pattern. PLS-DA models were validated using permutation tests based on the separation distance with 100 permutations. Significance cutoff for the non-hypothesis-based analysis was set as an FDR of <0.1 and, in the pathway analysis, with pathway impact of >0.1. Differences in NAD+-related metabolites and metabolite ratios, as well as the continuous normally-distributed baseline characteristics, were evaluated using Brown–Forsythe and Welch ANOVA with multiple comparisons with the *p*-value threshold set at 0.05. Categorical baseline characteristics were compared among groups using Fisher’s exact test. Urine specific gravity was compared among groups using the Kruskal–Wallis test with the *p*-value threshold set at 0.05. Urine hippurate was analyzed against KynA/Trp ratio using linear regression with the *p*-value threshold set at 0.05. One individual from Spain was excluded from the regression analysis as an outlier. Analyses were conducted using Metaboanalyst v. 5.0, Graphpad Prism v. 9, and R v. 4.1.0.

## 3. Results

Demographic and laboratory features of each group are reported in Table 2. A total of 136 metabolites were identified by ^1^H NMR and are reported in Appendix A, with 78 metabolites included in the final analysis after excluding metabolites with >50% of values below the threshold for detection.

### 3.1. Non-Hypothesis-Based Explorations

Comparisons in the urine metabolome across groups are shown in Figure 2. PLS-DA models used were all validated by permutation tests (Appendix A). Random forest was able to correctly distinguish workers in the CS group and the OW group with 9% (11/117 CS workers predicted to be OW) and 38% (30/78 OW workers predicted to be CS) misclassification, respectively (overall OOB error 21%); the CS group and the SW group with 3% and 8% misclassification, respectively (overall OOB error 5%); and the CS group and the NW group with 10% and 16% misclassification, respectively (overall OOB error 13%). Metabolic pathways differing across all three comparisons with FDR < 0.1 and pathway impact >0.1 were phenylalanine, glyoxylate/dicarboxylate, and glycine/serine/threonine metabolism.

Urinary creatinine-corrected hippurate was significantly higher in the CS group than in all other groups (mean mmol/mg creatinine [95% CI]: 0.85 [0.74, 0.96] in the CS group, 0.22 [0.18, 0.27] in the OW group, 0.22 [0.15, 0.28] in the SW group, 0.20 [0.18, 0.23] in the NW group; one-way ANOVA, FDR < 0.0001, *p* < 0.0001; Figure 3).

Comparisons in the urine metabolome across groups when pooled by risk factor are shown in Figure 4. Random forest distinguished individuals residing in Nicaragua, designated as the high-risk region, and Spain, designated as the low-risk region, with <1% and 46% misclassification, respectively (overall OOB error 10%); workers in high-intensity vs. lower intensity labor groups with 46% and 4% misclassification, respectively (overall OOB error 17%); and agricultural and non-agricultural workers with 1% and 89% misclassification, respectively (overall OOB error 25%).

### 3.2. Hypothesis-Based Explorations

Levels of the metabolites NAD+, Trp, KynA, Nam, and MNA compared among groups are shown in Table 3. Comparison between the CS and OW and SW and NW groups reveals higher levels of NAD+ and KynA, although differences were not significant across all pairwise comparisons.

Table 3 also shows relative levels of metabolites related to the de novo and salvage pathways of NAD+ metabolism. NAD+/Trp ratio was significantly higher in the CS group than in all other groups. KynA/Trp ratio was also higher in the CS group than in all other groups, although the difference was not significant in all pairwise comparisons. MNA/NAM ratio and NAD+/MNA ratio significantly differed among groups. NAD/MNA trended highest in the CS group, although it was significantly higher than only the OW group.

Post-hoc, we analyzed the relationship between the urinary hippurate and Kyn/Trp ratio, which was significantly and positively correlated (log(KynA/Trp) = 0.205 * [log(Hip)] + 0.705, 95% CI of slope = 0.119 to 0.291, *p* < 0.0001; Figure 5).

### 3.3. Stability Analyses

Results of heatmaps and mean decrease accuracy assessment from random forest comparisons across groups are shown in Appendix A and corroborate patterns described here. When we compared metabolites related to NAD+ metabolism, along with hippurate, among occupational groups with sugarcane harvest cutters and seed cutters analyzed separately (Appendix A), seed cutters were either similar to harvest cutters or fell between harvest cutters and the lower-risk referent groups.

When we repeated non-hypothesis based PLS-DA and pathway analyses in subset of individuals with an eGFR ≥ 90 mL/min/1.73 m^2^, findings were largely similar to the original analysis for the CS vs. OW and CS vs. NW comparisons (Appendix A). In the CS vs. SW comparison, the list of important metabolites was also similar, but proline betaine replaced hippurate as the metabolite with the highest VIP score and betaine emerged as an important metabolite.

## 4. Discussion

In this cross-sectional analysis of workers undertaking manual labor in Nicaragua and Spain, we observed that distinct urinary metabolic features distinguished our pre-specified high-risk group, Nicaraguan sugarcane harvest and seed cutters, from other occupational groups with respect to both the urinary metabolome as a whole and NAD+ metabolism in particular. 

One of the most striking differences between the high-risk CS group and other occupational groups was the degree of urinary hippurate elevation observed. Elevated hippurate may reflect increased gut permeability in the high-risk group. Prior study of the urine metabolome of heat-stressed goats demonstrated that elevated hippurate levels are a fundamental determinant of between-group differences, in conjunction with increases in other gut-related metabolites [32]. Our study also identified that increased levels of other gut-derived uremic metabolites, including 1-methylguanidine, 2-hydroxyphenylacetate, neopterin, and theobromine, were included among features important in distinguishing the CS group from referent groups. Elevated levels of these gut-related metabolites are in keeping with a proposed mechanism that increased gut permeability in response to heat stress may be a driver of MeN through increased inflammation [33]. 

Complementing these findings, in hypothesis-based analyses urinary KynA/Trp was increased in the CS group compared to referent groups, again consistent with a greater inflammatory state associated with high-risk work [18,34]. Prior studies have demonstrated increased Trp catabolism in association with intense exercise, with increased serum KynA/Trp ratio correlating with increased markers of inflammation [34,35,36]. Increased urinary values of KynA and KynA/Trp are observed in patients with AKI and are correlated with injury status [19]. The positive association observed between urinary hippurate and Kyn/Trp further supports the potential connection between increased gut permeability and inflammation in this population. 

Increased urinary hippurate could reflect other processes as well. Increased urinary hippurate is associated with increased renal blood flow [37]. Urinary hippurate is also affected by consumption of hippurate and its precursors. Of particular relevance to this study is bolis, the sugary rehydration packets sometimes used by sugarcane workers, that contain benzoic sodium which is metabolized to hippurate [38,39]. Benzoic sodium intake may also interact with Trp metabolism because it can be converted to anthranilic acid, an intermediary in the de novo pathway [38]. 

In our investigation of the NAD+ biosynthetic derangement, urinary NAD+ trended higher, rather than lower, in the high-risk CS occupational group, and urinary NAD+/Trp was increased rather than decreased. Unfortunately, the urinary Q/T ratio, a better validated marker of NAD+ biosynthetic derangement [10,11], was not calculable because quinolinate was not detected in the majority of specimens using our methodology. Moreover, NAD+ measurable in the urine has been shown to reflect extracellular immunomodulation rather than the altered intracellular energy metabolism characteristic of kidney injury [40,41]. Additionally, NAD+ can be sensitive to sample handling techniques, and measurement error may exist given specimens used in this study were not collected and processed specifically for NAD+ analysis. Further exploration of NAD+ biosynthetic derangement, including measurement of urinary quinolinate, is needed in populations at risk for MeN to draw any definitive conclusions regarding its role in disease pathogenesis. 

Patterns in TCA cycle intermediates, which tended to be lower in the CS group, further support the conclusion that NAD+ elevations in this study may not be a product of increased central energy metabolism. We observed decreased metabolites related to alanine/aspartate/glutamate metabolism, which is involved in the catabolism of proteins for energy generation by cellular mitochondria as an alternative input to the TCA cycle [42]. Similarly, glycine/serine/threonine metabolism and TCA—related metabolites themselves were generally decreased in the high-risk group. Creatine, which facilitates recycling of ATP, was also decreased in the high-risk group [43]. These findings are consistent with the overall impairment in central energy metabolism reported under states of injury and oxidative stress in the kidney [41].

TCA cycle components, such as citrate, 2-oxoglutarate, and fumarate are dicarboxylates, so decreased levels could alternatively reflect differences in sodium-dependent proximal tubular reabsorption, with increased sodium reabsorption associated with increased dicarboxylate reabsorption largely mediated through the sodium-dicarboxylate cotransporter NaDC1 [44]. Given MeN appears to be a tubulointerstitial kidney disease [2], further exploration of alterations in renal proximal tubular handling of metabolites in this population is merited. 

Analysis of the urine metabolome enabled us to distinguish the high-risk CS group from other groups with reasonable accuracy using random forest. This accuracy was generally lower when comparison groups were pooled to reflect shared risk based on region of residence, labor intensity, and work in agriculture. This suggests that metabolic elements defining the CS group are more distinct than metabolic elements defining any one shared risk factor. Increased hippurate as well as decreased TCA cycle intermediaries remained important features distinguishing groups, while features related to consumption, such as caffeine and proline betaine were key features distinguishing Spain and Nicaragua-based groups [45]. Overall, therefore, we were not able to make any strong conclusions about whether region of residence, labor intensity, or work in agriculture was the risk factor most likely to drive disease pathophysiology. 

Differences existed between the CS group and referent groups beyond the designed differences in work intensity, residence, and an agricultural work environment. These should be considered when assessing the findings presented here. The SW and NW groups were older on average than the CS group. Unlike for the CS group, specimens for the NW group were not universally collected at the end of a work day. The SW group also had worse kidney function on average than the CS group, and included more individuals with proteinuria and self-reported hypertension and fewer individuals with weekly NSAID use.

Another important consideration in interpreting the results of this study is that differing urinary metabolite levels may result from alterations in metabolism, alterations in transport from intracellular to extracellular spaces, and alterations in the kidney’s handling of those metabolites, all of which we are unable to differentiate between due to our observational, cross-sectional design. We did not evaluate how strenuous each individual’s labor was on the day of specimen collection, and it likely differed among individuals within the same occupational groups, potentially blurring the distinction in intensity of manual labor among groups. All groups engaged in some amount of manual labor, further limiting differences in this exposure between comparators; the NW group was composed of brickmakers and miners, both occupational groups that are also considered to be at increased risk for MeN [26,27]. Participants were not fasting or on a standardized diet, and dietary differences are likely an unmeasured confounder contributing to metabolite differences between groups. Samples were spot, rather than 24-h, urine samples, which enables assessment of metabolism at a point in time of interest but which in turn requires correction for variation in urine concentration. In our study we standardized metabolites to urinary creatinine [46]. A concern could be that more intense work results in increased endogenous creatinine production from muscle breakdown, affecting the utility of creatinine as a standard, although labor appears not to significantly impact urinary creatinine excretion [47]. Additionally, participants were excluded who had elevated serum creatinine at the time of urine specimen collection, which should further minimize variation in daily creatinine excretion. Specimens analyzed were not freshly obtained but had been frozen at −80 °C for up to five years. Nevertheless, storage under these conditions has been shown to effectively maintain the integrity of urinary metabolites [48]. Sample sizes were relatively small and geographically limited, and may not reflect other populations at risk for MeN.

Our findings suggest several areas for further exploration. These include future reevaluation of these study participants to determine who went on to develop chronic kidney disease, which can serve as a crucial additional comparison to understand whether certain metabolic features may either predispose to subsequent CKD development, or be biomarkers of early disease prior to decline in glomerular filtration rates. In addition, evaluating pre-, during-, and post-work specimens from high-risk groups would enable a more comprehensive determination of metabolic changes arising over the course of a work shift.

## 5. Conclusions

Features in the urinary metabolome of Nicaraguan sugarcane harvest and seed cutters differentiated them from other groups of workers performing manual labor. A key distinguishing factor was their elevated levels of hippurate and other gut-derived uremic metabolites. In addition, sugarcane harvest and seed cutters showed evidence of increased KynA generation, which may indicate the presence of a greater inflammatory state in these individuals. Taken together, our findings support the hypothesis that inflammation from increased gut permeability due to heat stress may be present in this high-risk occupational group. Other pathways distinguishing cane cutters and seeders from referent groups suggest there may be further physiologic differences, including alterations in NAD+ metabolism, central energy metabolism, and proximal tubular transport function. This study is among the first to date to explore the urinary metabolome among individuals at risk for MeN, and demonstrates the potential of metabolomics as a tool to explore the pathophysiology of this still incompletely understood disease.

## Figures and Tables

**Figure 1 metabolites-13-00325-f001:**
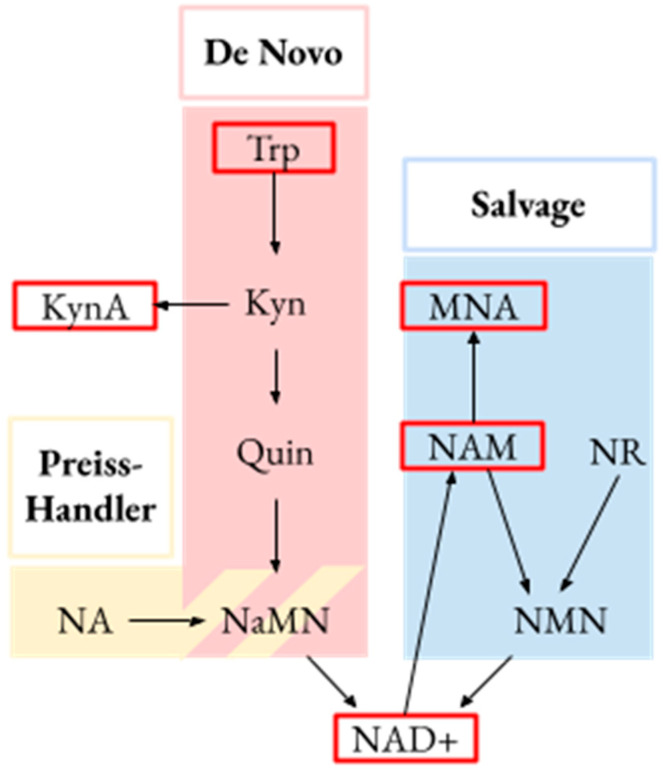
Key metabolic pathways connected to NAD+. De novo from tryptophan (Trp), salvage from nicotinate riboside (NR) and nicotinamide (NAM), and Preiss–Handler from nicotinic acid. Metabolites evaluated in this study are identified by a red border. Kyn, kynurenine; KynA, kynurenic acid; NaMN, nicotinate mononucleotide; NA, nicotinic acid; NAD+, nicotinamide adenine dinucleotide; MNA, methylnicotinamide; NMN, nicotinamide mononucleoside.

**Figure 2 metabolites-13-00325-f002:**
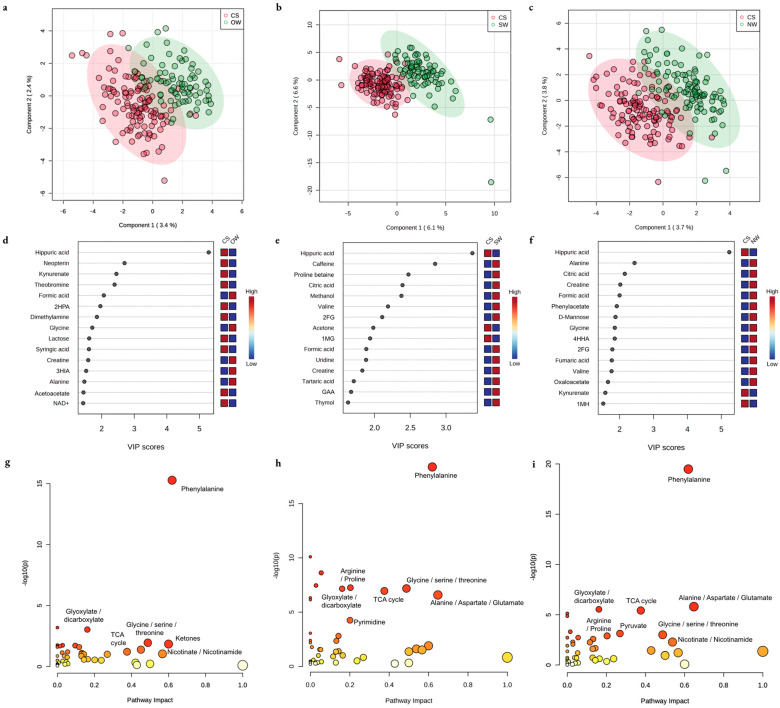
Differential metabolic features of urine between comparison groups using partial least squares discriminant analysis (PLS-DA). Two-dimensional visualization of the separation of groups by PLS-DA is shown in Nicaraguan cane harvest and seed cutters (CS) compared with (**a**) other cane workers (OW) in Nicaragua, (**b**) Spain agricultural workers (SW), and (**c**) non-agricultural workers (NW) in Nicaragua. Important features by variable importance in prognosis (VIP) score in PLS-DA are shown in the CS group compared to the (**d**) OW, (**e**) SW, and (**f**) NW groups. Significant metabolic pathways differentiating groups are shown in the CS group compared to the (**g**) OW, (**h**) SW, and (**i**) NW groups. 2HPA, 2-hydroxyphenylacetic acid; 3HIA, 3-hydroxyisovaleric acid; NAD+, nicotinamide adenine dinucleotide; 2FG, 2-furoylglycine; 1MG, 1-methylguanidine; GAA, guanidinoacetic acid; 4HHA, 4-hydroxyhippuric acid; 1MH, 1-methylhistidine; TCA, tricarboxylic acid.

**Figure 3 metabolites-13-00325-f003:**
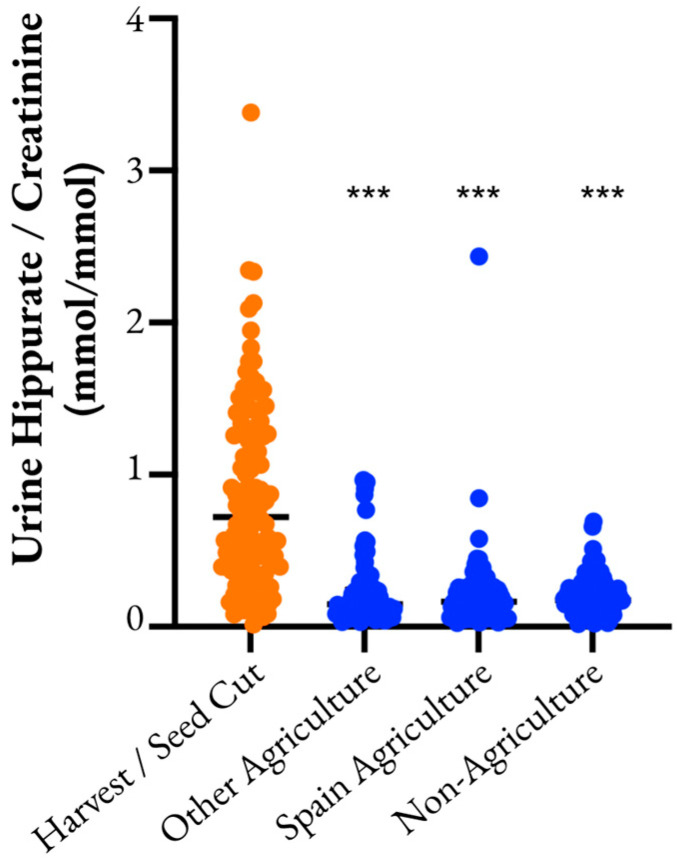
Urinary hippurate across labor-based comparison groups. Groups considered to have a greater risk for subsequent development of Mesoamerican nephropathy are in orange, and those with less risk are in blue. Hippurate is corrected for urinary creatinine level. *** Indicates significant difference from the harvest/seed cutter group at *p* < 0.0001.

**Figure 4 metabolites-13-00325-f004:**
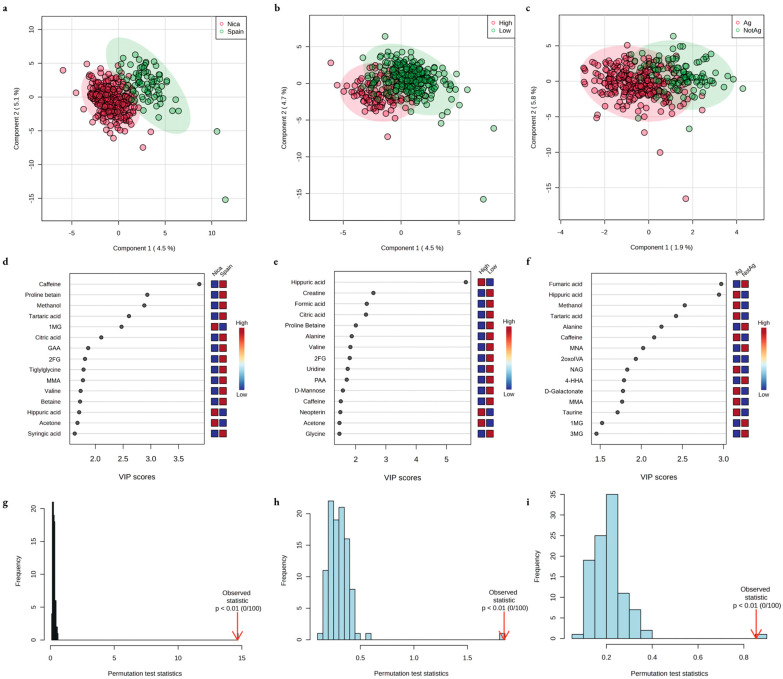
Differential metabolic features of urine between groups pooled by risk factor using partial least squares discriminant analysis (PLS-DA). Two-dimensional visualization of separation of groups by PLS-DA is shown in (**a**) individuals living in Nicaragua (Nica) vs. Spain, (**b**) individuals working in high- vs. low-intensity manual labor jobs, and (**c**) individuals working in agriculture (Ag) vs. not in agriculture (NotAg). Important features by variable importance in prognosis (VIP) score in PLS-DA are shown for (**d**) Nicaragua vs. Spain, (**e**) high- vs. low-intensity labor, and (**f**) agriculture vs. not in agriculture. Permutation test validation based on separation distance is shown for PLS-DA models for (**g**) Nicaragua vs. Spain, (**h**) high- vs. low-intensity labor, and (**i**) agriculture vs. not in agriculture. 1MG, 1, methylguanidine; GAA, guanidinoacetic acid; 2FG, 2-furoylglycine; MMA, methylmalonic acid; PAA, phenylacetic acid; MNA, 1-methylnicotinamide; 2oxoIVA, 2-oxoisovaleric acid; NAG, N-acetylglutamate; 4-HHA, 4-hydroxyhippuric acid; 1MG, 1-methylguanidine; 3MG, 3-methylglutaconic acid.

**Figure 5 metabolites-13-00325-f005:**
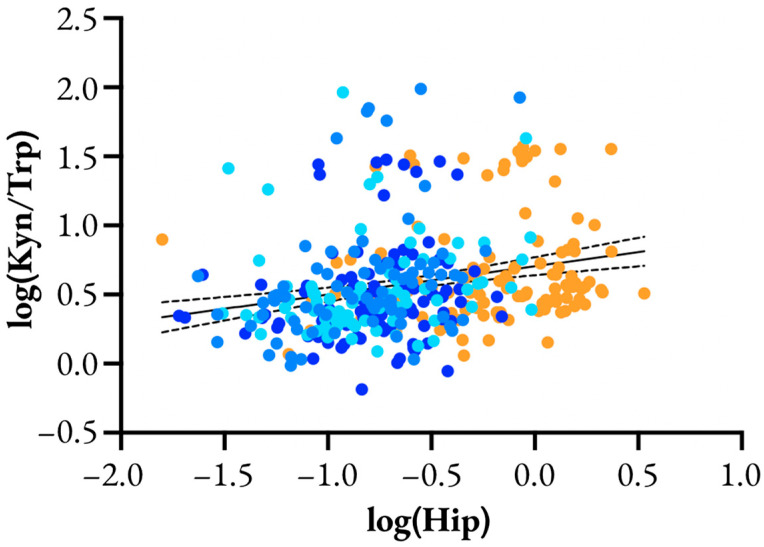
Log-log plot of urine hippurate (Hip) vs. kynurenic acid to tryptophan ratio (KynA/Trp). Different occupational groups are represented by color: •, cane harvest and seed cutters; •, other cane workers; •, Spain agricultural workers; •, non-agricultural workers. Solid line with dashed error represents the linear regression model with 95% confidence intervals.

**Table 1 metabolites-13-00325-t001:** Population-level pre-specified risk factor exposure assumptions. Groups hypothesized to have greater risk for subsequent development of Mesoamerican nephropathy are in orange, and those with less likelihood are in blue. “+” indicates a high association with each risk factor, “−” indicates a lower association with each risk factor.

Group	Residence in a High-Risk Region	Highest-Intensity Manual Labor	Agricultural Environment
Cane Harvest and Seed Cutters	+	+	+
Other Cane Workers	+	−	+
Spain Agricultural Workers	−	−	+
Non-Agricultural Workers	+	−	−

**Table 2 metabolites-13-00325-t002:** Baseline and basic laboratory features of the study populations. * indicates a difference in pairwise comparison from the cane harvest and seed cutters group. ^+^ indicates that data were derived only from the brickmaker subgroup as they were not available from the miner subgroup. BMI, body mass index; eGFR, estimated glomerular filtration rate; ACEi, angiotensin converting enzyme inhibitor; ARB, angiotensin receptor blocker; NSAID, non-steroidal anti-inflammatory drug.

Variable	Cane Harvest and Seed Cutters *n* = 117	Other Cane Workers *n* = 78	Spain Agricultural Workers *n* = 78	Non-Agricultural Workers *n* = 102	*p*-Value
Age in years, mean (SD)	35 (8)	36 (9)	42 (11) *	48 (9) *	<0.0001
Weight in kg, mean (SD)	67 (11)	75 (11) *	80 (13) *	79 (16) *	<0.0001
Height in cm, mean (SD)	168 (9)	169 (7)	172 (7) *	167 (7) ^+^	0.0005
BMI in kg/m^2^, mean (SD)	25 (4)	26 (4)	27 (4) *	25 (4) ^+^	0.006
Serum creatinine in mg/dL, mean (SD)	0.9 (0.1)	0.8 (0.1)	1.0 (0.1) *	0.8 (0.1)	<0.0001
eGFR, mL/min/1.73 m^2^, mean (SD)	112 (12)	114 (11)	95 (13) *	106 (12) *	<0.0001
Urine creatinine in mmol/L, mean (SD)	132 (86)	126 (67)	157 (67)	140 (73)	0.049
Urine dipstick specific gravity, median (IQR)	1.015 (1.010 to 1.020)	1.015 (1.010 to 1.020)	1.030 (1.020 to 1.030) *	1.020 (1.015 to 1.020) ^+^	<0.0001
Urine dipstick protein, *n* (%)					
0–30 mg/dL	117 (100)	78 (100)	68 (87)	100 (98)	<0.0001
>30–<300 mg/dL	0 (0)	0 (0)	5 (6)	2 (2)	
≥300 mg/dL	0 (0)	0 (0)	5 (6)	0 (0)	
Urine dipstick leukocyte esterase > trace, *n* (%)	12 (10)	4 (5)	5 (6)	1 (1)	0.02
Hypertension, *n* (%)	8 (7)	2 (3)	16 (21)	10 (10)	0.002
Diabetes, *n* (%)	0 (0)	0 (0)	3 (4)	0 (0)	0.02
Current use of ACEi or ARB, *n* (%)	0 (0)	0 (0)	5 (6)	0 (0)	0.0007
NSAID use category, *n* (%)					
<1×/month	76 (65)	51 (65)	44 (56)	70 (69)	<0.0001
1×/month to 1×/week	21 (18)	12 (15)	30 (38)	25 (25)	
>1×/week	20 (17)	15 (19)	4 (5)	7 (7)	
Tobacco use category, *n* (%)					
Current smoker	51 (44)	23 (29)	27 (35)	38 (37)	0.54
Former smoker	21 (18)	20 (26)	19 (24)	25 (25)	
Never smoker	45 (38)	35 (45)	32 (41)	39 (38)	
Alcohol > 1×/month, *n* (%)	52 (47)	26 (34)	36 (46)	15 (68) ^+^	0.04
Years working in occupational category, mean (SD)	12 (5)	15 (6)	14 (11)	14 (11)	0.11

**Table 3 metabolites-13-00325-t003:** Key urinary metabolites of NAD+, tryptophan, and nicotinamide metabolism. Groups hypothesized to be at greater risk of metabolic derangement along these metabolic pathways are orange, at less risk blue. * indicates a statistically significant difference from the cane harvest/seed cutters group at a *p*-value of <0.05, ** at <0.01. NAD+, nicotinamide adenine dinucleotide; Cr, urine creatinine; IQR, interquartile range.

	Cane Harvest and Seed Cutters (*n* = 117)	Other Cane Workers (*n* = 78)	Spain Agricultural Workers (*n* = 78)	Non-Agriculture (*n* = 102)	*p*
**NAD+**, mmol/mmol Cr					
median	0.0156	0.0126 **	0.0122 **	0.0146	0.0004
IQR	0.0120, 0.0197	0.0094, 0.0164	0.0093, 0.0181	0.0124, 0.0185	
**Tryptophan**, mmol/mmol Cr					
median	0.0056	0.0055	0.0061	0.0068 **	0.01
IQR	0.0038, 0.0072	0.0042, 0.0075	0.0043, 0.0094	0.0043, 0.0093	
**Kynurenic acid**, mmol/mmol Cr					
median	0.0200	0.0160 **	0.0184	0.0188 *	<0.0001
IQR	0.0166, 0.0249	0.0140, 0.0189	0.0160, 0.0255	0.0152, 0.0217	
**Nicotinamide**, mmol/mmol Cr					
median	0.0141	0.0130	0.0139	0.0126	0.22
IQR	0.0111, 0.0181	0.0112, 0.0182	0.0104, 0.0260	0.0101, 0.0167	
**Methylnicotinamide**, mmol/mmol Cr					
median	0.0040	0.0039	0.0035	0.0048 *	0.0004
IQR	0.0030, 0.0055	0.0031, 0.0057	0.0027, 0.0047	0.0034, 0.0069	
**DE NOVO PATHWAY**					
**NAD+ to Tryptophan Ratio**, mmol/mmol					
median	2.92	2.32 *	2.12 *	2.06 **	0.006
IQR	1.99, 5.23	1.53, 4.24	1.32, 4.89	1.43, 4.49	
**Kynurenic Acid to Tryptophan Ratio**, mmol/mmol					
median	3.48	2.89 *	3.06	2.66 **	0.001
IQR	2.61, 6.34	2.12, 3.91	2.15, 4.88	1.95, 4.19	
**NAD+ to Kynurenic Acid Ratio**, mmol/mmol					
median	0.727	0.698	0.594	0.758	0.01
IQR	0.542, 1.057	0.561, 0.896	0.462, 1.138	0.610, 0.990	
**SALVAGE PATHWAY**					
**NAD+ to Nicotinamide Ratio**, mmol/mmol					
median	1.022	0.939	0.924	1.193	0.07
IQR	0.765, 1.597	0.684, 1.260	0.351, 1.796	0.773, 1.627	
**Methylnicotinamide to Nicotinamide Ratio**, mmol/mmol					
median	0.304	0.301	0.236	0.374 *	<0.0001
IQR	0.204, 0.407	0.193, 0.500	0.110, 0.346	0.248, 0.575	
**NAD+ to Methylnicotinamide Ratio**, mmol/mmol					
median	3.68	2.72 **	3.52	3.01	0.007
IQR	2.72, 5.89	2.17, 3.93	2.47, 5.31	2.07, 4.90	

## Data Availability

Not applicable.

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
