# Peer review of "Metabolic Features of Increased Gut Permeability, Inflammation, and Altered Energy Metabolism Distinguish Agricultural Workers at Risk for Mesoamerican Nephropathy"

_metabolites, 2023, doi:10.3390/metabo13030325_

Round 1
Reviewer 1 Report
The manuscript "Metabolic Features of Increased Gut Permeability, Inflammation, and Altered Energy Metabolism Distinguish Agricultural Workers at Risk for Mesoamerican Nephropathy" tried to compare the relative amounts of metabolites in urine of various participants regarding their activity. Despite its interest this study has majors concerns.
First, regarding clinical data it missed important informations. The wheigt and height of the participant are missing. What is the definition of hypertension? Same for diabetes? Other drugs used must be stated also with tobacco, alcool and illicite drugs in table 1. Exposure to pollutants must be stated. It missed important data regarding the urine with basic results like ionogram, urea, creatinine and proteinuria. This is very important to have an idea of the urine concentration and the kidney response at the time of the collection.
Second, regarding the metabolites measurments . Why the authors excluded the metabolites with more than 50% of values below threshold? I don’t understand why methionine was not identified in the urine of patient same remark for the L-kynurenine or cystine? Some explanation must be provide why some metabolites are not present, the absence is bias or not. Is the same proportion of missing metabolites regarding the various groups? It is very important to have this basic data.
Third, regarding ethical issues. The authors gave no data about the consent to participate of the participants of the study. We need to know if basic ethic rules were followed in Nicaragua. The other ethical consents is regarding the authors, despite the fact that more than two third of participants are from Nicaragua, no authors from this country is present. How the authors had access to sample? If no nicaraguan authors is present, it look like scientific coloniasm.
Fourth, the absence of quinolinate to perform Q/T ratio is really problematic to interpret the results.
Minor remarks, in the abstract, “Nicaraguans doing non-agricultural work 35 (n=107)” in the whole manuscript the number of this group is 102
Reviewer 2 Report
Thank you for the opportunity to review the article “Metabolic Features of Increased Gut permeability: inflammation and altered energy metabolism distinguish agricultural workers at risk for Mesoamerican Nephropathy (MeN)”. Besides several limitations of cross-sectional observational studies, this study gives us exciting insights into possible pathogenetic mechanisms of MeN. The authors, using proton nuclear magnetic resonance (1H NMR), investigated urine metabolites at the end of a work shift among three agricultural groups (Nicaraguan CS, OW, and Spanish Workers) and one non-agricultural worker. The study is focused on nocitinamide adenine dinucletide (NAD+) related metabolites and showed increased kynurenate/tryptophane ratio in the high-risk group. The paper underlies the role of a high inflammatory state, partly explained by gut permeability and altered energy metabolism rather than repetitive episodes of AKI, in the pathogenesis of MeN. Furthermore, the study demonstrates some potential biomarkers that may finally lead to the development of unreversible kidney damage - chronic kidney disease (CKD) in agriculture communities in the Pacific coastal region of Mesoamerica.
The study design and all its items are written according to the STROBE statement, and no revision is needed.
There are several potential sources of bias in this study: small and geographically limited sample size, poor description of the metabolic state of individuals of each group, leading to different transport mechanisms between intra- and extracellular spaces; there is unclear the kidney’s handling of various metabolites in single participant; no standardized diet was used; samples were not freshly obtained.
Besides all mentioned sources of bias, this observational cross-sectional study is a gateway for discovering potential biomarkers, like Hippurate and urinary kynurenic acid, predicting the development of MeN in a high-risk population and deserves publication.
Reviewer 3 Report
The paper by Raines et al addresses the problem of Mesoamerican Nephropathy. The discovery of risk factors for this disease is very important and needed. The introduction provides insufficient background. Methods are not adequately described. My main concerns are with the research design and interpretations of the results. The compared groups are very poorly described, there is no justification why exactly such groups were singled out and in what respect OW, SW, and NW groups are reference groups.
My specific comments are listed below:
- Line 30: it seems that the word “syndrome” does not fit in this context.
- Line 36: NW group size is different in the abstract and manuscript
- Line 71: in ref 7 brown adipose tissue and muscle were analyzed, how it translates to use this reference in the context of the “organ function”?
- Lines 66-71: lack of references
- Line 115: why was such a value of eGFR adopted? Depending on the age such eGFR may suggest kidney damage with mild loss of function, however, the authors wrote that “All individuals in this study had preserved kidney function”. What was the eGFR for each group? Was it comparable?
- Line 122: Why were samples from NW taken at a different time? "At different time " means at what time? it was a different time for each sample? doesn't a different time interfere with the ability to compare samples?
- Line 129: On what criteria were the samples selected for the NW group? NW counts 102 samples and in each of the quoted studies, there were many more participants. The inclusion criteria for the analysis should be very clearly specified. There is no information in the paper that the NW group is based on a Nicaraguan Mining Community and workers employed by brickmaking facilities therefore occupations are also associated with a high risk of MeN
- Lines 138-142: How this passage relates to the materials and methods section?
- Results: no data on education, smoking status, BMI, water consumption, history of CKD in the family, hours worked per week, proteinuria
- Table 2: Where is the SD for age? Where is the SD for creatinine? no statistical analysis of whether the variables differ significantly between the analyzed groups. Hypertension and diabetes are separate risk factors for CKD, does the inclusion of such patients in the study not interfere with the analysis? Why did the authors decide to include only patients on anti-inflammatory drugs? Why is there no information about this anywhere in the main text? Doesn't taking anti-inflammatory drugs to interfere with the assessment of inflammation metabolic features?
Doesn't the fact that there was a percentage of hypertensive patients in each group but only one group used the drugs interfere with the analysis? Were hypertension and diabetes self-reported or confirmed by medical data? No explanation of the used abbreviations.
- Figure 2: unreadable font too small
- Figure 3: unit missing
- Lines 242-248: If the results are important enough to be discussed in detail in the main text, then why is the figure in the supplement?
- Line 252-253: “ higher levels of NAD+ and KynA, and lower levels of Trp in the CS group, as well as higher levels of MNA in the NW group” – there is no statically significant difference in NAD between CS and NW, there is no statically significant difference in KynA between CS and SW, there is no statically significant difference in Trp between CS and OW, CS and SW, and many more in this paragraph
- Table 2: What does “Cr” mean?
Round 2
Reviewer 3 Report
- The reviewer has no further comment